# Satellite measurements of peroxyacetyl nitrate from the Cross-Track Infrared Sounder: Comparison with ATom aircraft measurements

Vivienne H. Payne[1], Susan S. Kulawik[2], Emily V. Fischer[3], Jared F. Brewer[3,4], L. Gregory Huey[5], Kazuyuki Miyazaki[1], John R. Worden[1], Kevin. W. Bowman[1], Eric J. Hintsa[6, 7], Fred Moore[6, 7], James W. Elkins[7] and Julieta Juncosa Calahorrano[3]

[1]Jet Propulsion Laboratory, California Institute of Technology, Pasadena, California, USA
[2]Bay Area Environmental Research Institute / NASA Ames, Mountain View, California, USA
[3]Colorado State University, Fort Collins, Colorado, USA
[4]Harvard University, Cambridge, Massachusetts, USA
[5]Georgia Tech, Georgia, USA
[6] Cooperative Institute for Research in Environmental Sciences, University of Colorado Boulder, Boulder, Colorado, USA
[7] Global Monitoring Laboratory, NOAA, Boulder, Colorado, USA

*Correspondence to*: Vivienne H. Payne (vivienne.h.payne@jpl.nasa.gov)

**Abstract.** We present an overview of an optimal estimation algorithm to retrieve peroxyacetyl nitrate (PAN) from single field of view Level 1B radiances measured by the Cross-Track Infrared Sounder (CrIS). CrIS PAN retrievals show peak sensitivity in the mid-troposphere, with degrees of freedom for signal less than or equal to 1.0. We show comparisons with two sets of aircraft measurements from the Atmospheric Tomography Mission (ATom), the PAN and Trace Hydrohalocarbon ExpeRiment (PANTHER) and the Georgia Tech Chemical Ionization Mass Spectrometer (GT-CIMS). We find a systematic difference between the two aircraft datasets, with vertically averaged mid-tropospheric values from the GT-CIMS around 14 % lower than equivalent values from PANTHER. However, the two sets of aircraft measurements are strongly correlated ($R^2$ value of 0.92) and do provide a consistent view of the large-scale variation of PAN. We demonstrate that the retrievals of PAN from CrIS show skill in measurement of these large-scale PAN distributions in the remote mid-troposphere compared to the retrieval prior. The standard deviation of individual CrIS-aircraft differences is 0.08 ppbv, which we take as an estimate of the uncertainty of the CrIS mid-tropospheric PAN for a single satellite field of view. The standard deviation of the CrIS-aircraft comparisons for averaged CrIS retrievals (median of 20 satellite co-incidences with each aircraft profile) is lower, at 0.05 ppbv. This would suggest that the retrieval error reduces with averaging, although not with the square root of the number of observations. We find a negative bias of order 0.1 ppbv in the CrIS PAN results with respect to the aircraft measurements. This bias shows a dependence on column water vapor. We provide a water vapor dependent bias correction for use with the CrIS PAN data.

## 1 Introduction

Peroxyacetyl nitrate (PAN) is formed through the oxidation of volatile organic compounds (VOCs) in the presence of nitrogen oxide radicals ($NO_x$) (Roberts, 2007). PAN formation and decomposition provide important pathways by which $NO_x$ emissions are redistributed (Singh and Hanst, 1981; Singh et al., 1986) and contribute to downwind oxidant formation (Wang et al., 1998). PAN is a particularly difficult compound to capture and validate in models because many factors impact the production and lifetime of this species (Fischer et al., 2014). PAN has a low background abundance, is often a clear tracer of photochemistry, and its abundance can be highly variable in space and time. Therefore, it can be difficult to know whether the limited in-situ measurements are broadly representative. Aside from a few exceptions (e.g., Fiore et al., 2018; Pollack et al., 2013) PAN is not routinely measured in surface air quality networks. It is often only measured *in situ* as part of relatively short (i.e., weeks to months) field campaigns (e.g., Alvarado et al., 2010; Fischer et al., 2010; Juncosa Calahoranno et al., 2020), aimed at elucidating specific chemical process. Thus existing *in situ* observations provide snapshots of this species, they cannot be used to probe changes over time, and they offer a limited view of the spatial distribution.

Recent work on ground-based remote sensing of PAN from stations of the Network for the Detection of Atmospheric Composition Change (NDACC) provides the promise of long-term measurements, albeit with limited spatial coverage (Mahieu et al., 2021). Satellite remote sensing provides a means for observations over long timescales with global coverage. While limb-sounding satellite observations can provide global-scale information on PAN in the upper troposphere and lower stratosphere with high vertical resolution and sensitivity (Glatthor et al., 2007; Moore and Remedios, 2010; Wiegele et al., 2012; Tereszchuk et al., 2013; Pope et al., 2016; Ungermann et al., 2016), nadir-viewing satellite observations can offer sensitivity to PAN variations lower in the troposphere. Observations and retrievals of PAN have previously been reported from the Tropospheric Emission Spectrometer (TES) (Alvarado et al., 2011; Payne et al., 2014) and from the Infrared Atmospheric Sounding Interferometer (IASI) (Clarisse et al., 2011; Franco et al., 2018). Nadir-viewing observations have shown large enhancements in PAN associated with fires and have been used to shed new light on the role of fires, PAN precursor emissions and dynamics on the global distribution of PAN and on long-range transport of ozone (Zhu et al., 2015, 2017; Payne et al., 2017; Jiang et al., 2016; Fischer et al., 2018).

Here we show new retrievals of PAN from the Cross-Track Infrared Sounder (CrIS). We demonstrate the capability of CrIS to measure variations in background PAN levels over the remote ocean. Section 2 provides an overview of the satellite and aircraft measurements used in this work, while Section 3 describes the CrIS PAN retrieval algorithm. Results are presented in Section 4. Section 5 provides discussion of the results and conclusions.

## 2 Measurements

### 2.1 CrIS satellite radiances

The Cross-Track Infrared Sounder (CrIS) (Han et al., 2013) is a high spectral resolution spectroradiometer. CrIS instruments are currently flying on the Suomi National Polar-Orbiting Partnership (S-NPP) satellite and on the National Oceanic and Atmospheric Administration NOAA-20 satellite as part of the Joint Polar Satellite System (JPSS). CrIS instruments will be also included on the payload for 3 more JPSS satellites, with a plan to extend the CrIS record to 2035 and beyond. CrIS is a Fourier transform spectrometer and provides measurements of Earth view interferograms at 30 cross-track positions, each with a 3 x 3 array of field of views (FOVs). The diameter of the FOVs is 15 km at nadir. The interferograms are processed to provide calibrated and geolocated Level 1B spectra in three bands: 660-1095 cm$^{-1}$ (longwave), 1210-1750 cm$^{-1}$ (mid-wave) and 2155-2550 cm$^{-1}$ (shortwave). The full spectral resolution radiances are supplied on a 0.625 cm$^{-1}$ spectral grid.

In this work, we use radiances from S-NPP CrIS, although the algorithm described here can be applied to data from any of the CrIS instruments. We use NASA version 2 Level 1B radiances (Revercomb and Strow, 2018) from the Goddard Earth Sciences Data and Information Services Center (GES DISC). S-NPP flies in a sun-synchronous orbit with a mean local daytime overpass time of 13:30. Radiometric calibration is described in Tobin et al. (2013) while the noise characteristics are described in Zavyalov et al. (2013). The CrIS noise is low compared to other high resolution thermal infrared sounders, such as the TES, IASI, and the Atmospheric Infrared Sounder (AIRS). This low noise, combined with the afternoon orbit, enables good sensitivity to a range of trace gases (e.g. Shephard and Cady-Pereira., 2015; Fu et al., 2019).

### 2.2 ATom aircraft measurements

The Atmospheric Tomography Mission (ATom) was a series of aircraft campaigns to study the impact of human-produced air pollution on greenhouse gases and chemically reactive gases in the atmosphere (Wofsy et al., 2018). The mission consisted of four campaigns, covering four seasons: ATom-1 (July-August 2016), ATom-2 (January-February 2017), ATom-3 (September-October 2017) and ATom-4 (April-May 2018). An extensive payload was deployed on the NASA DC-8 aircraft for global-scale sampling of the atmosphere, with flight tracks involving continuous profiling between 0.2 and 12 km altitude. Figure 1 shows the locations of 500 mbar points for aircraft profiles flown on these 4 campaigns. The majority of these profiles are located over remote ocean.

There were two different PAN instruments flown on the ATom campaigns. The Georgia Tech Chemical Ionization Mass Spectrometer (GT-CIMS) (Huey et al., 2007) was flown on ATom-2, ATom-3 and ATom-4, but not on the ATom-1 campaign. The PAN and Trace Hydrohalocarbon ExpeRiment (PANTHER) (Elkins et al., 2001; Wofsy et al., 2011) uses electron capture detection and gas chromatography to measure PAN and was flown on ATom-1, -2, -3 and -4. Figure 2 and Figure 3 show "curtains" of PAN profile measurements from the two instruments. For the purposes of these figures, the aircraft profiles have

been split by longitude into "Pacific" (Region 1, longitude < -60º) and "Atlantic" (Region 2, longitude > -60º). It can be seen from this figure that the two instruments show a consistent picture of the large-scale PAN distribution. However, there are

some differences between the PAN measurements from the two aircraft instruments, with PANTHER values systematically higher than those from the GT-CIMS.  These differences are discussed further in Section 4.

## 3 CrIS PAN retrievals

### 3.1 Retrieval algorithm and strategy

The single footprint CrIS PAN retrievals shown in this work were produced using the MUlti-SpEctra, MUlti-SpEcies, MUlti-

Sensors (MUSES) retrieval algorithm (Fu et al., 2013, 2016, 2018; Worden et al., 2019). MUSES utilizes an optimal estimation approach with a priori constraints (Rodgers, 2000) and has heritage in the TES retrieval algorithm (Bowman et al. 2006). The forward model used within MUSES for this work is the Optimal Spectral Sampling (OSS) fast radiative transfer model (Moncet et al., 2008, 2015). We use OSS v1.2, trained using the Line By Line Radiative Transfer Model (LBLRTM) v12.4 (Clough et al., 2005; Alvarado et al., 2013).

Provided that the retrieved state is close to the true state, the retrieved state can be expressed as:

$$\hat{\mathbf{x}} = \mathbf{x}_a + \mathbf{A}(\mathbf{x} - \mathbf{x}_a) + \mathbf{Gn} + \mathbf{GK}_b(\mathbf{b} - \mathbf{b}_a) + \mathbf{G\Delta f} , \quad (1)$$

where $\hat{\mathbf{x}}$, $\mathbf{x}_a$, and $\mathbf{x}$ are the retrieved, a priori, and the "true" state vectors.  For the TES trace gas retrievals, the state vectors

were expressed as the natural logarithm of volume mixing ratio (VMR). The gain matrix, $\mathbf{G}$, maps from radiance space into profile space.  The vector $\mathbf{n}$ is the noise on the spectral radiances.  The vector $\mathbf{b}$ represents the true state for parameters that affect the modelled radiance but are not included in the retrieval state vector (such as calibration, concentrations of interfering gases, etc.). The vector $\mathbf{b}_a$ holds the corresponding a priori values. The Jacobian, $\mathbf{K}_b = \partial \mathbf{L}/\partial \mathbf{b}$ , describes the sensitivity of the forward modelled radiances $\mathbf{L}$ to the vector $\mathbf{b}$. The vector $\mathbf{\Delta f}$ represents the error in the forward model relative to the true

physics. Spectroscopic errors would be one component of the forward model error.

The averaging kernel, $\mathbf{A}$, describes the sensitivity of the retrieved state to the true state:

$$\mathbf{A} = \frac{\delta \hat{\mathbf{x}}}{\delta \mathbf{x}} = (\mathbf{K}^T \mathbf{S}_n^{-1} \mathbf{K} + \mathbf{R})^{-1} \mathbf{K}^T \mathbf{S}_n^{-1} \mathbf{K} = \mathbf{GK} \qquad (2)$$

Here,       $\mathbf{K}$   is   the   sensitivity   of   the   forward   modeled   radiances   to   the   state   vector $(\mathbf{K} = \frac{\delta \mathbf{L}}{\delta \hat{\mathbf{x}}})$.  The noise covariance matrix, $\mathbf{S}_{n,}$ represents the noise in the measured radiances.  $\mathbf{R}$ is the constraint matrix for the retrieval.

For profile retrievals, the widths of the rows of **A** provide a measure of the vertical resolution of the retrieval. Provided that the retrieval is relatively linear, the sum of each row of **A** indicates the fraction of retrieval information that comes from the measurement as opposed to the a priori at a given altitude (Rodgers, 2000). ("Relatively linear" means that although the retrieval problem itself is non-linear (and requires iteration to reach a solution), a linearization about some prior state is adequate to find a solution.) The trace of the averaging kernel matrix gives the number of degrees of freedom for signal (DOFS), or independent pieces of information, for the retrieval.

We use the PAN feature centred around 790 $cm^{-1}$. Figure 4 shows the PAN signal as seen in CrIS brightness temperatures for an example case from ATom-1. The CrIS PAN retrievals are performed in linear, rather than logarithmic volume mixing ratio (vmr). This figure shows a wide wavenumber range, but the chosen microwindows for the PAN retrievals are only a small subset of this. The microwindows used in the PAN retrievals are highlighted in red in Figure 4(b). We used PAN cross sections taken from the work of Allen et al. (2005a; 2005b), which cover a temperature range of 250-295 K. Allen et al. (2005a) cite uncertainties in the integrated band intensity of around 7 % for the band used here, but this does not include the extrapolation error below 250 K. Following the logic of Tereszchuk et al. (2013), we assume a value of ~13 % for the spectroscopic error. The main interferents overlapping with the PAN feature are water vapor ($H_2O$), carbon dioxide ($CO_2$), ozone ($O_3$) and carbon tetrachloride ($CCl_4$). The retrieval windows have been chosen to avoid the strong $H_2O$ line at 784.5 $cm^{-1}$, the complex of strong $H_2O$ lines at 790-801 $cm^{-1}$, the $CO_2$ Q-branch at 791.5 $cm^{-1}$ and the peak of the $CCl_4$ absorption. PAN retrievals are performed after previous steps to fit surface and atmospheric temperature, surface emissivity, $H_2O$, $O_3$ and cloud optical depth and cloud top pressure.

CrIS PAN retrievals are being processed routinely under the NASA Tropospheric Ozone and Precursors from Earth System Sounding (TROPESS) project and are publicly available via the GES DISC. The TROPESS datasets at the GES DISC include the forward stream (Bowman, 2021), where data are processed with low latency using the latest stable version of the algorithm that is available at the date of the satellite measurement. (There are also plans to release a "reanalysis" dataset, where the long-term record will be processed with a uniform algorithm version.) The TROPESS CrIS forward stream data are subsampled using a grid sampling approach where the region is divided into 0.8 degree lat x 0.8 degree lon grid boxes and the single, center-most target within each box is selected to be included in the dataset. The forward stream dataset provides both day and night time coverage. The TROPESS datasets also include so-called "special collections" where the sampling may be tailored to address a particular scientific study (or studies). Data for this work were processed specifically for matches with ATom aircraft profiles (see Section 4).

Figure 5 shows a global map of CrIS PAN for an example day, using the grid box subsampling referred to above. The map shows colored points for retrievals that pass quality screening. Retrievals were not processed for latitudes south of 70S.

There are gaps in coverage associated with strongly cloudy regions. There are also noteable gaps in coverage over desert areas. These areas are screened out due to the presence of a strong silicate feature in the surface emissivity spectrum that happens to coincide with the 790 cm$^{-1}$ PAN feature.

Figure 5 also shows regions of persistent negative PAN values over ocean in the Tropics. With retrievals in linear vmr, we would expect some negative values in the distribution retrieved values under low PAN conditions that would go negative.

However, ideally the mean over a large number of low PAN retrievals would be slightly positive overall. Figure 6(a) shows a scatter plot of retrievals by latitude, with a running mean, showing overall negative values in zonal means in the tropics. (We also see negative zonal means at for high southern latitudes.) The negative values in the tropics are discussed further in Section 4.

### 3.2 Initial guess and a priori constraints

The CrIS PAN retrievals presented here use the same a priori constraint vectors as the TES PAN retrievals, as described in Payne et al. (2014). We use a monthly-varying prior constraint, constructed using a GEOS-Chem v9.01.01 global chemical transport model simulation, as described in detail in Fischer et al. (2014). There are six possible constraint vectors for any given month, based on whether the location is in or outside the Tropics and whether the model predicts "clean", 'enhanced, maximum at surface" or "enhanced, maximum aloft" for a given location.  The a priori constraint matrix differs from that

described in Payne et al. (2014), since the TES v7 PAN retrieval is performed in ln(vmr). The constraint is loose in the troposphere and tighter in the stratosphere. We have chosen to use a diagonal constraint. For nadir retrievals, where vertical information is limited, it is common practice to introduce off-diagonal elements in the constraint matrix in order to avoid spurious oscillations in the retrieved profile. However, for fitting the broad PAN spectral feature, we did not find it necessary to introduce off-diagonal elements in the constraint matrix.  The initial guess profile values for these CrIS PAN retrievals are

set to a vanishingly small number.

### 3.2 Vertical sensitivity

The CrIS PAN retrievals are primarily sensitive to variations in PAN in the free troposphere, with peak sensitivity around 400-500 hPa. An example averaging kernel is shown in Figure 6(b). The DOFS for the CrIS PAN retrievals are generally less than 1.0, which means that the retrievals do not provide information on the vertical distribution of PAN. The latitudinal distribution

of the DOFS for an example day are shown in Figure 6(c). The highest values of DOFS are for clear-sky conditions. The presence of clouds will lead to reduced (or elimination of, depending on the cloud optical depth) sensitivity  to PAN below the cloud. We choose to retrieve PAN on multiple levels in order to preserve information that the averaging kernels provide about vertical sensitivity for individual soundings, but collapse to a single quantity per sounding for the purposes of presenting retrieved PAN values. Results are presented here in terms of a pressure-weighted average of the retrieved PAN between ~800

hPa and ~300 hPa. (The pressures for the relevant retrieval levels are actually 825 and 316 hPa). It can be seen from Figure

6(b) that we do expect the CrIS PAN retrievals to be sensitive to variations in PAN above 300 hPa. The choice of the 800 hPa to 300 hPa range here is a trade-off between the vertical range of the sensitivity of the retrieval and the vertical coverage of the aircraft measurements available for validation. The partial DOFS in this range is shown in Figure 6(d).

## 4 Results

The averaging kernels for the CrIS PAN retrieval are broad in vertical extent. Therefore, for the comparisons between CrIS and ATom PAN observations, we selected only the aircraft profiles that span at least the 800 to 300 hPa pressure range. GEOS-Chem model output for runs specific to the time period was appended above the uppermost and below the lowermost altitudes spanned by these ATom PAN profiles. The GEOS-Chem runs used for this purpose were version 12.0.0, 2x2.5 degree resolution, GEOS-FP meteorology. GEOS-Chem 12.0.0 uses the Community Emissions Database (CEDS; Hoesly et al., 2018)

as a global base case for anthropogenic emissions, overwritten with a series of local emissions inventories where appropriate: in Asia (MIX; Li et al., 2017), the United States (2011 NEI; US EPA, 2015), Europe (EMEP; EMEP, 2015; Vestreng & Klein, 2002), and Africa (DICE; Marais & Wiedinmyer, 2016). Biomass burning emissions are from the Global Fire Emissions Database (GFED) version 4 (Giglio et al., 2013), while biogenic emissions are from the Model of Emissions of Gases and Aerosols from Nature (MEGAN) V2.1 (Guenther et al., 2012).


We used coincidence criteria of 9 hours and 50 km to match CrIS FOVs to the aircraft profiles. The CrIS PAN retrievals were screened to exclude soundings with poor fits to observed radiances and those that did not pass quality control for the water vapor retrieval step (performed before the PAN step). In addition, we only included cases here where there were at least 5 CrIS FOVs that match with a given aircraft profile and pass the retrieval quality screening. For the CrIS-PANTHER comparisons,

337 aircraft profiles were considered in total for all four campaigns, with the number of good CrIS matches per aircraft profile ranging from 5 to 72 (median of 20). For the CrIS-GTCIMS comparisons, 239 aircraft profiles were considered in total for the three available campaigns, with the number of good CrIS matches per aircraft profile ranging from 5 to 67 (median of 20).

For each of the aircraft profiles, we applied the retrieval prior and averaging kernel for each of the individual matched CrIS

FOVs to the appended aircraft profiles to calculate a "convolved aircraft profile" for each FOV, representing what we would expect from the satellite retrieval if the appended aircraft profile represented the true atmospheric state viewed by the satellite. We then calculate a pressure-weighted average between 800 and 300 hPa for each convolved aircraft profile for comparison with the CrIS results.

A linear fit of the GT-CIMS-based vertical averages (from the "convolved" profiles) to those from PANTHER (for the 239 profiles where both datasets were available) results in a gradient of 0.86 and a negligible intercept. The $R^2$ value for the linear fit is 0.92, indicating that while there there is a systematic difference between the two sets of aircraft measurements,

they are very strongly correlated. The scatter around the fitted line for this comparison of GT-CIMS/PANTHER vertical averages is around 0.015 ppbv.


Figure 6 shows CrIS-PANTHER differences plotted by latitude for each of the 4 campaigns. A summary of the mean bias and standard deviation of CrIS-aircraft comparison results is provided in Table 1 for both PANTHER and GT-CIMS aircraft measurements. The standard deviation of the CrIS-aircraft comparisons for individual CrIS soundings is 0.08 ppbv (for both aircraft datasets). This suggests that 0.08 ppbv is a reasonable estimate for the uncertainty on a single CrIS PAN retrieval. The standard deviation of the CrIS-aircraft comparisons for averaged CrIS retrievals (multiple satellite co-incidences with each aircraft profile) is lower, at 0.05 ppbv. This would suggest that the retrieval error reduces with averaging, although not with the square root of the number of observations.

The optimal estimation approach used within the TROPESS CrIS PAN algorithm provides an estimate of the "noise-only" component of the observation error as part of the retrieval output. Comparisons between the values of this estimated observation error for individual soundings and the standard deviation of the CrIS-aircraft comparisons for individual CrIS soundings (empirical observation error) indicate that the estimated errors output from the optimal estimation retrieval are of the order of 0.03 ppbv, lower than the observed CrIS-aircraft standard deviations by a factor of 2-3. As described in Section 3.1, selected interferents are retrieved in previous retrieval steps before the PAN step. These retrieved interferents, while in the category of "systematic errors" would be expected to contribute a pseudo-random component to the error budget. For the purposes of estimating systematic error contributions, we performed a set of offline runs where the estimated retrieval errors for temperature, water vapor and ozone are propagated into the PAN retrieval. When the retrievals are run in this way, the mean estimated observation error for PAN is 0.06 ppbv, which is closer to, but still somewhat lower than, the standard deviation of the CrIS/aircraft differences. We find that the propagated temperature error is the dominant piece of this, followed by water vapor and that the error associated with ozone is negligible.

The next biggest interferent contribution in the PAN microwindows used in the retrieval is from CCl4, which is not retrieved. The current algorithm uses a CCl4 climatology that is static in time and was constructed around the time of the launch of the Aura satellite (2004). Ground-based measurements from the NOAA Global Monitoring Laboratory indicate that CCl4 has decreased by around 20 % between 2004 and the current day. There are plans to update this (and other) climatology(ies) in future algorithm versions to account for known time dependence. We performed runs where the CCl4 climatology was scaled by 0.8 in order to get a sense of the magnitude of systematic error associated with mis-specification of CCl4. The mean impact on the PAN retrievals for the set of ATom match-ups was only 0.01 ppbv for the 20 % perturbation. CCl4 is expected to continue to decrease over time, so if not addressed, this would become a bigger issue as time goes on.

The contributions to the error budget considered here are summarized in Table 2. The aggregated uncertainty of 0.09 is similar to the empirical value of 0.08 from the standard deviation of CrIS-aircraft comparisons for individual CrIS soundings. Since the uncertainties on the CrIS PAN retrievals are large compared to those on the aircraft measurements, the differences between PANTHER and GT-CIMS aircraft datasets do not affect the conclusions about the CrIS PAN retrieval uncertainties and the magnitude of the bias in the CrIS PAN.

Figure 7(a) shows the CrIS-PANTHER comparisons in a scatter plot. The CrIS points in Figure 8(a) correspond to the "averages for each aircraft profile" shown in Figure 7. The $R^2$ value between CrIS and PANTHER values  (all 4 campaigns) is 0.62. The $R^2$ value between the retrieval prior and PANTHER values is 0.38. The gradient of the linear fit between CrIS and aircraft values in Figure 8(a) is 1.07+/-0.06. A bootstrapping analysis with 10,000 samples indicates that this gradient and associated uncertainty are robust. The gradient of the linear fit between the prior and the aircraft values is 0.41. (For the GT-CIMS dataset from ATom 2, 3, and 4, the $R^2$ value between CrIS and aircraft is 0.64 and the $R^2$ value between prior and aircraft is 0.53.)

There is some additional component of uncertainty associated with what we have assumed for the profile above the top of the aircraft measurements. Due to the vertical sensitivity of the CrIS PAN retrieval, the assumption about what to append above the top of the aircraft profile is far more important than what is appended at the bottom. If targeted GEOS-Chem model runs for the campaign time period had not been available, an alternative crude approach could have been to simply append the retrieval prior to the top of the aircraft profiles. The difference between these two approaches provides some estimate of the uncertainty associated with the assumed profile above the top of the aircraft measurement. We find a 20 % reduction in the aircraft/satellite slope between the case where we append the prior and the case where we append the dedicated GEOS-Chem runs. We can think of this as a pessimistic estimate of the error associated with the assumption of the profile above the top of the aircraft profiles.

The results demonstrate overall consistency between CrIS-PANTHER results for the four separate campaigns and between the CrIS-PANTHER and CrIS-GTCIMS results. The CrIS PAN retrievals show skill relative to the monthly climatology used for the retrieval prior (increased $R^2$ value for a linear fit to the aircraft values) whether the PANTHER or GT-CIMS aircraft dataset is used. Nonetheless, there is an overall bias in the CrIS-aircraft comparisons, and there is some variation in this bias. We find that the bias depends on column water vapor.  Figure 9 shows scatter plots of CrIS/aircraft differences against column water vapor for the four campaigns. We performed some off-line diagnostic runs to allow us to examine the spectral residuals over a wider region surrounding the CrIS PAN mircrowindows, using the atmospheric and surface state that were fit in advance of the PAN step. We find that there is an overall positive offset in the radiance residuals between 760 and 775 $cm^{-1}$ before the PAN is retrieved. This can in fact be seen in the example shown in Figure 4. This offset tends to be larger for cases with larger column water vapor. This could indicate either a systematic bias in the retrieved water vapor that is fitted in steps before the

285 PAN step, or a systematic bias in some aspect of the water vapor spectroscopy in this specific spectral region. (Note that we do exclude cases where the quality flag for the water vapor retrieval from a previous step was "bad".) We speculate that if the true atmospheric PAN were uniform with latitude, with all values at the low end of the range (say < 0.1 ppbv), then we would see a retrieval bias that would follow the latitudinal structure of water vapor. At very low PAN values, the retrieval tends to compensate for the offset by underestimating the PAN. However, at higher PAN values, the shape of the PAN feature is

290 stronger, allowing the retrieval to better distinguish the shape of PAN from an offset and so the bias is reduced. This is consistent with a satellite/aircraft slope that is greater than 1.0. In future versions of the algorithm, we might hope to develop an approach to correct for this radiance offset before fitting the PAN. For the version of the CrIS PAN product that is already publicly available, we suggest that users add a water vapor dependent bias correction to the mid-tropospheric average PAN, $c = 0.05 + 0.035*10^{-23}*X$, where X is the column density of water vapor (in molecules/cm$^2$). This corresponds to the black line

shown in all four panels in Figure 9, which is the linear fit to data from all four ATom campaigns. Figure 8(b) shows a scatter plot of CrIS vs aircraft PAN after the application of this water vapor dependent bias correction. It can be seen that the correction removes the overall bias. The slope (1.05 compared to 1.07) and the correlation coefficient (0.66 compared to 0.62) are slightly improved after the application of the bias correction. The bias correction does not change the overall standard deviation in the differences between CrIS and aircraft, at least not within the number of significant figures cited in Table 1.

**Table 1. Mean and standard deviation of CrIS-aircraft differences, by instrument and campaign**

| ATom campaign (dates) | CrIS-PANTHER PAN [ppbv] | | | | CrIS-GTCIMS PAN [ppbv] | | | |
| --- | --- | --- | --- | --- | --- | --- | --- | --- |
| | Individual CrIS | | Averaged CrIS | | Individual CrIS | | Averaged CrIS | |
| | Mean | Std. dev. | Mean | Std. dev. | Mean | Std. dev. | Mean | Std. dev. |
| ATom 1 July-August 2016 | -0.06 | 0.09 | -0.09 | 0.06 | n/a | n/a | n/a | n/a |
| ATom 2 January-February 2017 | -0.09 | 0.08 | -0.09 | 0.04 | -0.08 | 0.08 | -0.08 | 0.04 |
| ATom 3 September-October 2017 | -0.06 | 0.07 | -0.06 | 0.04 | -0.06 | 0.08 | -0.06 | 0.05 |
| ATom 4 April-May 2018 | -0.09 | 0.08 | -0.09 | 0.05 | -0.07 | 0.09 | -0.08 | 0.06 |
| All | -0.08 | 0.08 | -0.08 | 0.05 | -0.07 | 0.08 | -0.07 | 0.05 |

**Table 2. Estimated contributions to the PAN retrieval error for a single CrIS sounding**

| Index | Uncertainty | Nature | Estimated magnitude for a case with 0.2 ppbv in mid troposphere | |
|---|---|---|---|---|
| | | | **ppbv** | **%** |
| 1 | Instrument noise | random | 0.03 | 15 % |
| 2 | Total calculated observation error, including propagation of temperature, water vapor, ozone retrieval errors | Pseudo-random | 0.06 | 30 % |
| 3 | Spectroscopic uncertainty | Systematic | 0.02 | 13 % |
| 4 | CCl$_4$ | Systematic | -0.01 or less | -5 % or less |
| | Aggregate of estimated uncertainties for retrieval over ocean | | 0.09 | 45 % |

## 5 Discussion and Conclusions

We have developed an algorithm for retrieval of PAN from single field of view L1B CrIS radiances and have validated results against two sets of aircraft profile measurements from ATom. The CrIS PAN retrievals are primarily sensitive in the mid-troposphere and have 1.0 or fewer DOFs, meaning that they do not provide information on the vertical distribution of PAN. We show results in terms of a single vertically averaged quantity. We find a negative bias of order 0.1 ppbv in the CrIS PAN results with respect to the aircraft measurements, with good consistency between results from the four different ATom campaigns. For the future, more validation data over land and over a wider range of PAN values would be desirable. For remote regions, the observed bias is large relative to absolute PAN values, although it would be a smaller fraction of the absolute PAN values in fire plumes, for example. The observed bias in the CrIS PAN shows dependence on water vapor, and we recommend that users apply the empirical water vapor dependent correction described in Section 4. CrIS single footprint water vapor retrieval products are available as part of the same TROPESS data collections as the CrIS PAN products described here. The results suggest a single sounding uncertainty of around 0.08 ppbv, and demonstrate the ability of the CrIS PAN retrievals to capture variation in the "background" PAN values observed over remote ocean regions from ATom. For demonstration of the capability of CrIS to capture PAN in fire plumes, we refer to recent work by Juncosa Calahorrano et al. (2021).

## Acknowledgements

This work was carried out at the Jet Propulsion Laboratory, California Institute of Technology, under a contract with the National Aeronautics and Space Administration (80NM0018D0004) . CrIS Level 1B radiances were accessed via the GES

DISC. We thank Steve Wofsy (Harvard University) and the ATom campaign team for all the effort that went into the campaigns and for making the datasets available. We wish to thank the Sounder SIPS team at JPL for assistance with obtaining the CrIS radiances, and the MUSES/TROPESS software and science teams for their support and insights. We also thank the NASA Sounder Science Team for helpful feedback and discussions.

**Data availability**

The ATom aircraft datasets were obtained from https://doi.org/10.3334/ORNLDAAC/1581 (Wofsy et al., 2018). MUSES-CrIS PAN products from S-NPP and JPSS-1 are available via the GES-DISC from the NASA Tropospheric Ozone and Precursors from Earth System Sounding (TROPESS) project at https://disc.gsfc.nasa.gov/datasets/TRPSDL2PANCRSFS_1/summary and https://disc.gsfc.nasa.gov/datasets/TRPSDL2PANCRS1FS_1/summary respectively. The CrIS – aircraft matched dataset

used here for validation is available from the authors on request.

**Author contributions**

VHP, SSK and EVF were responsible for study design. VHP and SSK were responsible for algorithm development, data analysis and manuscript writing. JFB was responsible for GEOS-Chem runs for the ATom time periods. LGH, SCW, JE, EH,

FM and EVF were responsible for providing aircraft measurements and guidance on their use. KM and JJC contributed to overall assessment of the CrIS data quality and manuscript editing. JRW contributed to the interpretation of validation results. JRW and KWB are responsible for data dissemination via management of the TROPESS project.

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

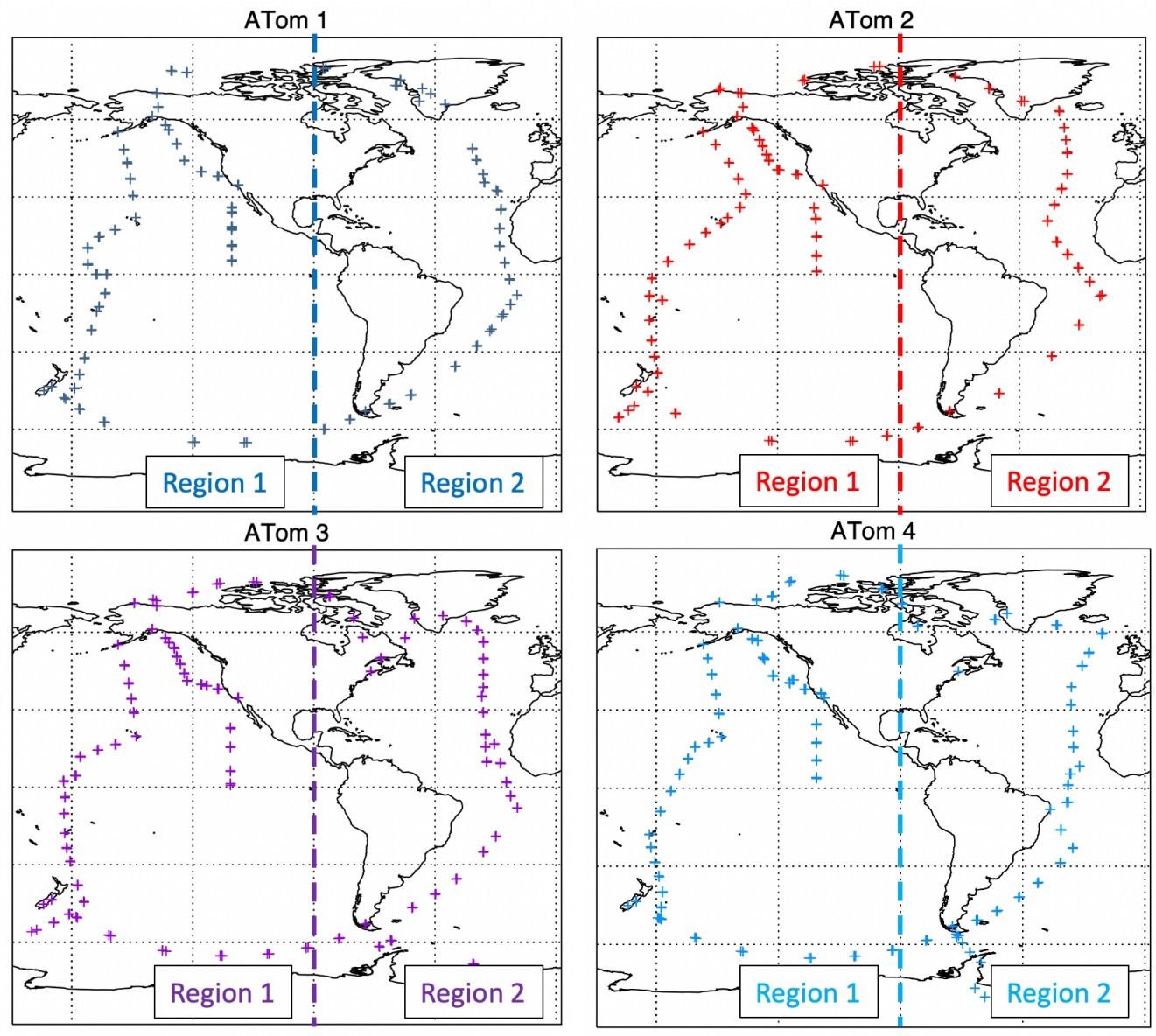

**Figure 1. Locations of 500 mbar pressure in aircraft profiles measured during the ATom campaigns.  The vertical line at 60 degrees shows the separation into "Region 1" (Pacific) and "Region 2" (Atlantic).**

## Region 1 (Pacific)

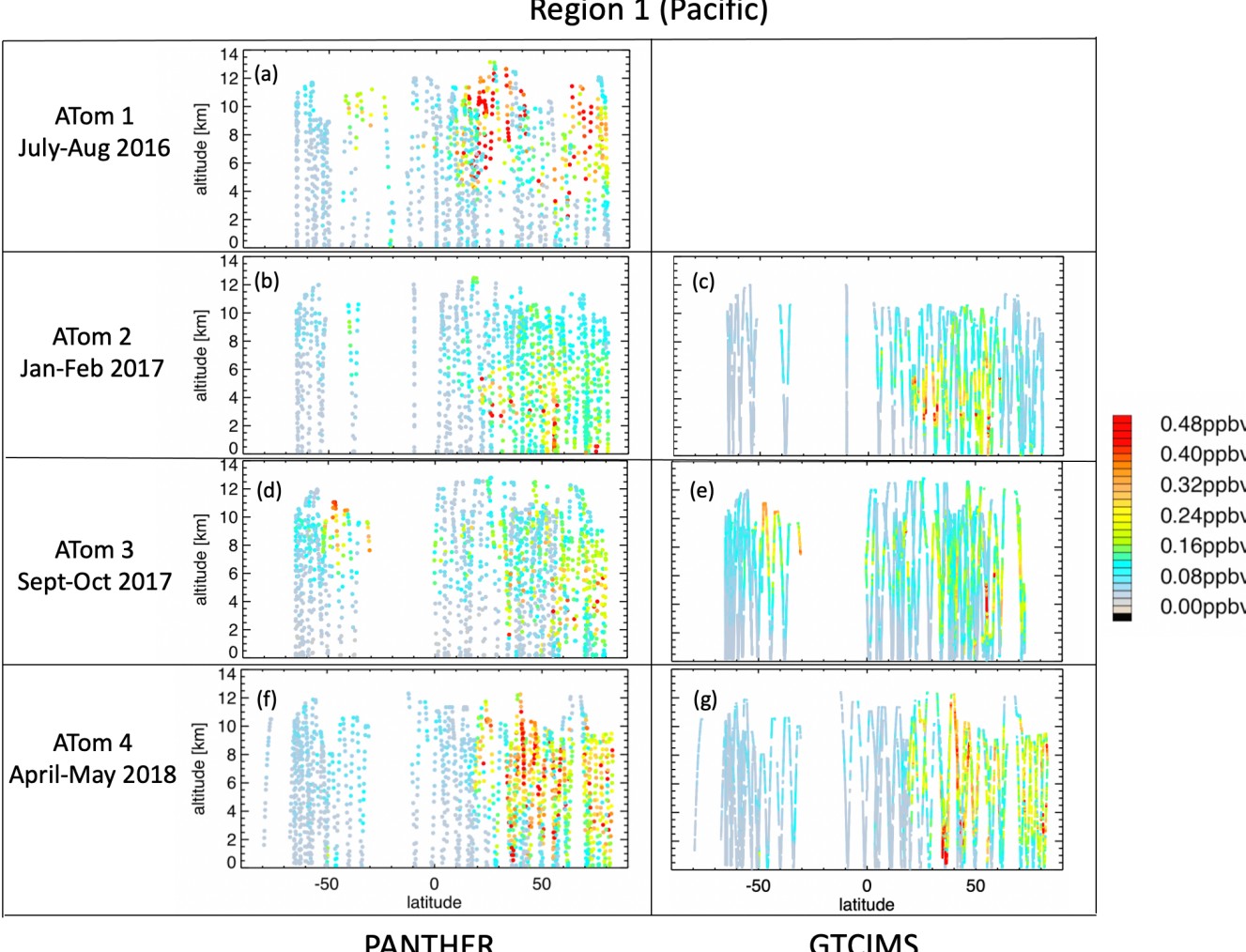

PANTHER             GTCIMS

**Figure 2. Aircraft measurements of PAN from PANTHER (a, b, d, f) and GTCIMS (c, e, g) instruments from ATom 1, 2, 3 and 4 for Region 1 ("Pacific", longitude less than -60º).**

# Region 2 (Atlantic)

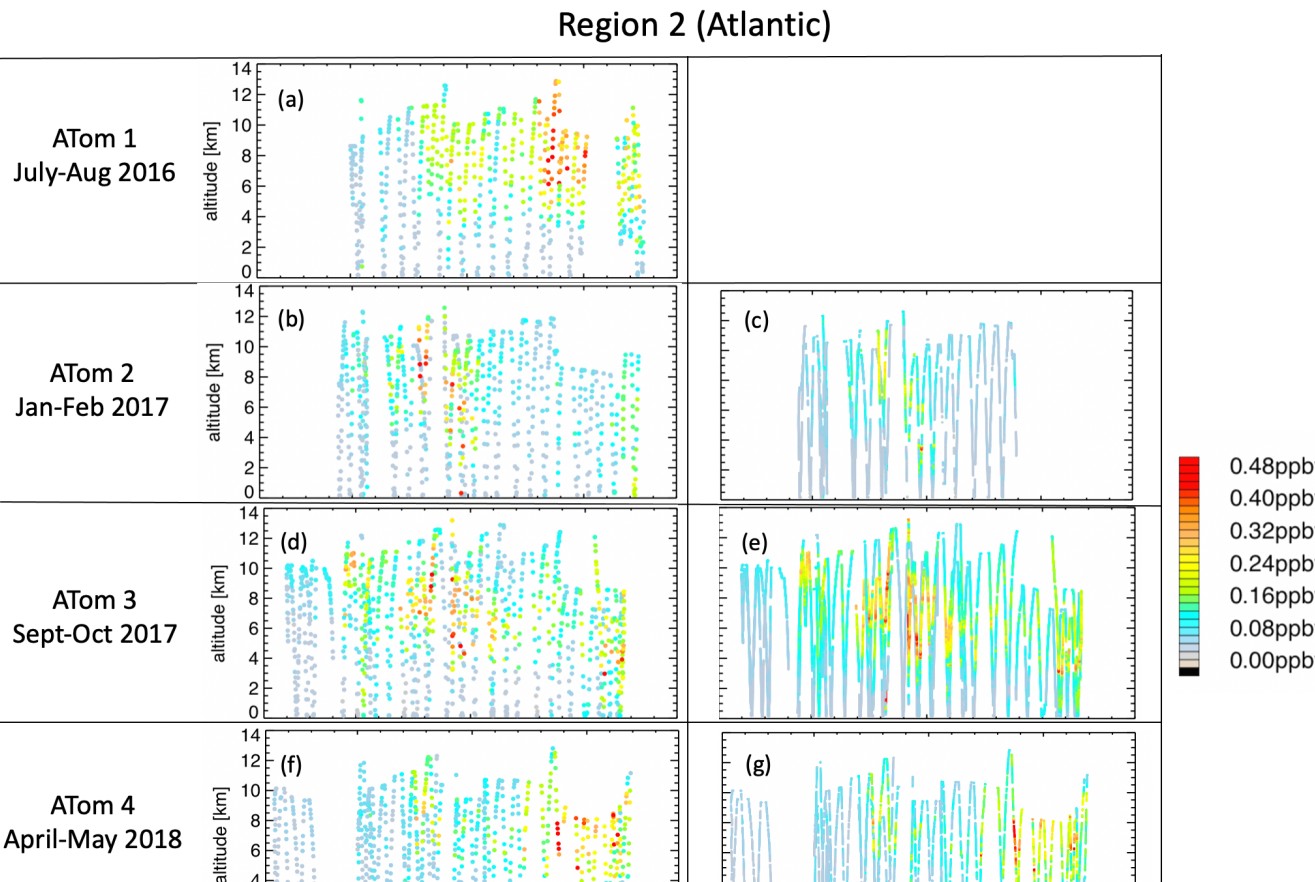

**Figure 3. Aircraft measurements of PAN from PANTHER (a, b, d, f) and GTCIMS (c, e, g) instruments from ATom 1, 2, 3 and 4 for Region 1 ("Atlantic", longitude greater than -60º).**

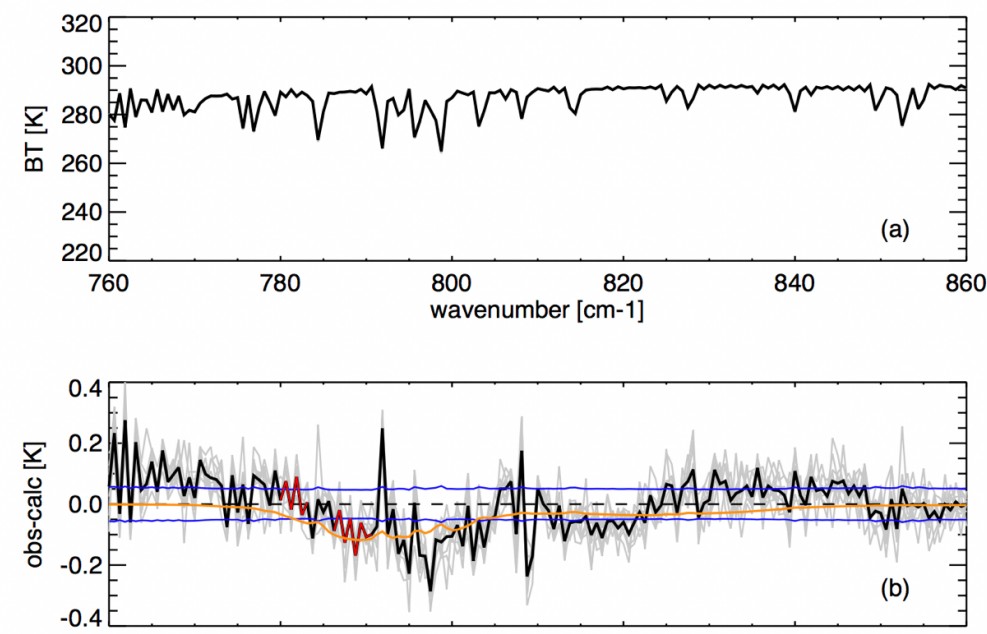

**Figure 4: (a) Measured CrIS brightness temperature spectrum for a case co-located with an ATom-1 profile on 17th August 2016, at 36.2 degrees latitude, -27.3 degrees longitude. (b) Residual spectra (observed – calculated) for match-ups with this ATom-1 case, with zero PAN in the calculation, after retrieval of surface and atmospheric temperature, water vapor, cloud optical depth, cloud top pressure and surface emissivity. Grey lines show residuals for individual CrIS FOVs that meet the co-incidence criteria. Black line shows the mean residual for this set. Red segments show the microwindows used in the retrievals. Blue lines show the CrIS instrument noise (noise equivalent delta brightness temperature, or NEdT) for a single CrIS FOV. Orange line shows the shape of the PAN spectral feature.**


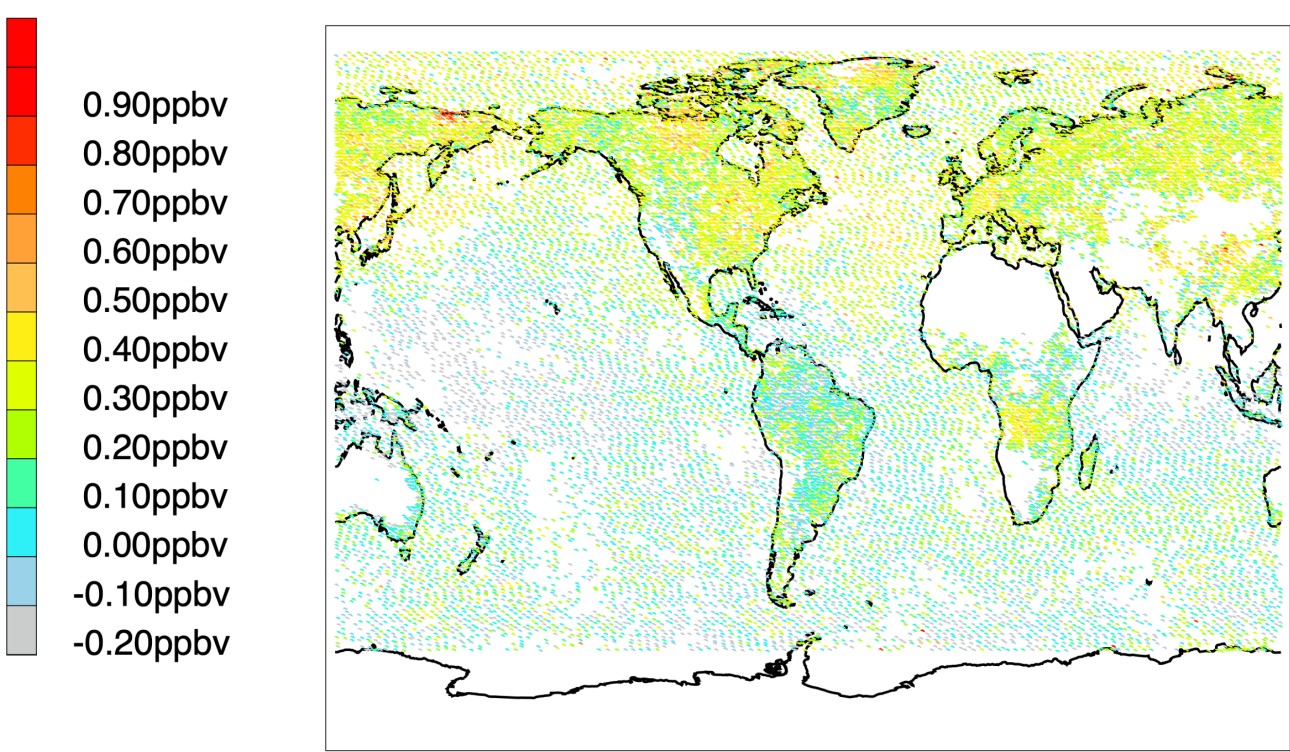

**Figure 5. CrIS PAN for 1st August 2020 (sub-sampled by selecting the center-most CrIS FOV in each 0.8 by 0.8 degree grid box for latitudes above 70S). Note that the gray color includes points where the CrIS PAN is below -0.20 ppbv.**




Figure 6. (a) CrIS PAN retrievals for sub-sampled measurements for 1st August 2020, plotted by latitude. (b) Averaging kernel for an example mid-latitude case (from a point at the center of the black diamond shown in (a), (c) and (d). Panels (c) and (d) show total and partial (825-316 hPa) DOFS for retrievals from this day. In all panels, the black solid line shows a running mean over a 200 point window.

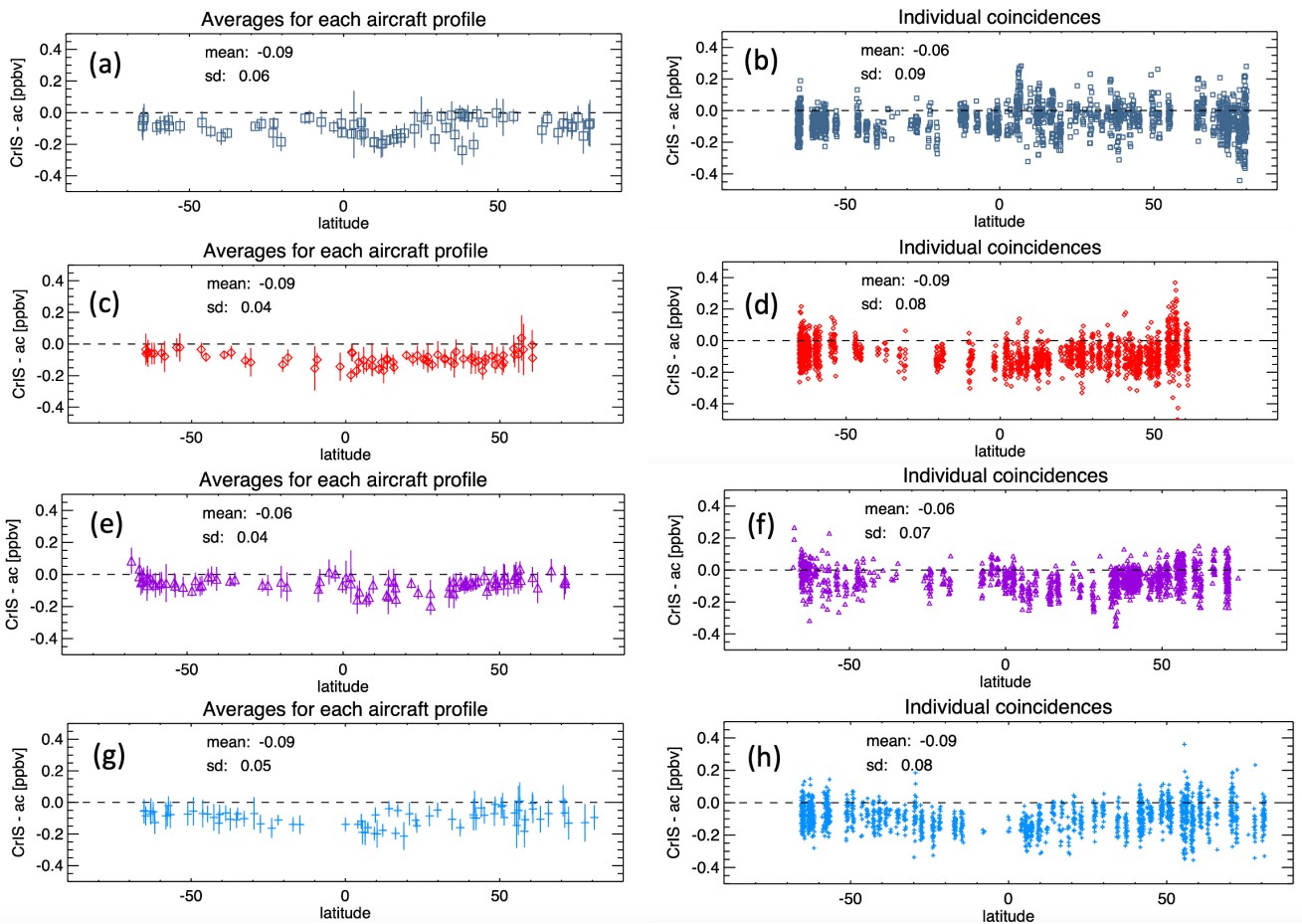

**Figure 7. CrIS-PANTHER PAN differences against latitude for the four ATom aircraft campaigns. Results from ATom 1 are shown in panels (a) and (b), ATom 2 in (c) and (d), ATom 3 in (e) and (f) and ATom 4 in (g) and (h). Panels (a), (c), (e) and (g) show the results after averaging all the CrIS co-incidences for each aircraft profile. Vertical bars on the points in these panels show the standard deviation of the averaged CrIS result for each aircraft profile. Panels (b), (d), f) and (h) show the results for individual CrIS soundings (multiple CrIS co-incidences with each aircraft profile.**


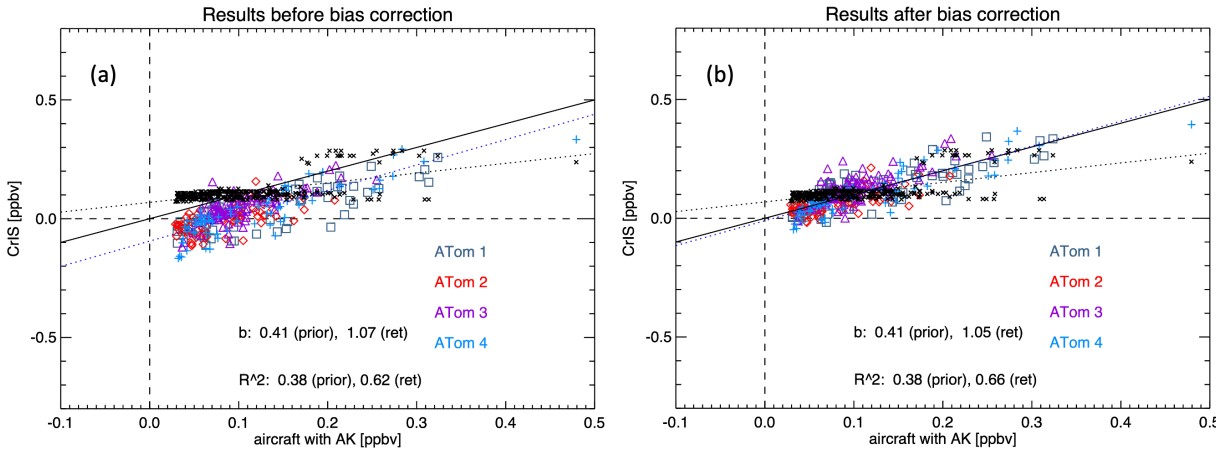


**Figure 8. Scatter plots of CrIS PAN against PANTHER aircraft measurements, (a) before and (b) after application of a water vapor dependent bias correction. Colored symbols show CrIS retrieval results, while the black 'x's show the prior values used in the CrIS retrievals. CrIS results shown here are averages of the CrIS co-incidences for each of the aircraft profiles. The 1:1 line is shows as solid black, while the dashed blue and black lines show the linear fit for the CrIS retrieved and prior results respectively. Gradients (b-values) of the linear fits and R² values are shown on the figure.**


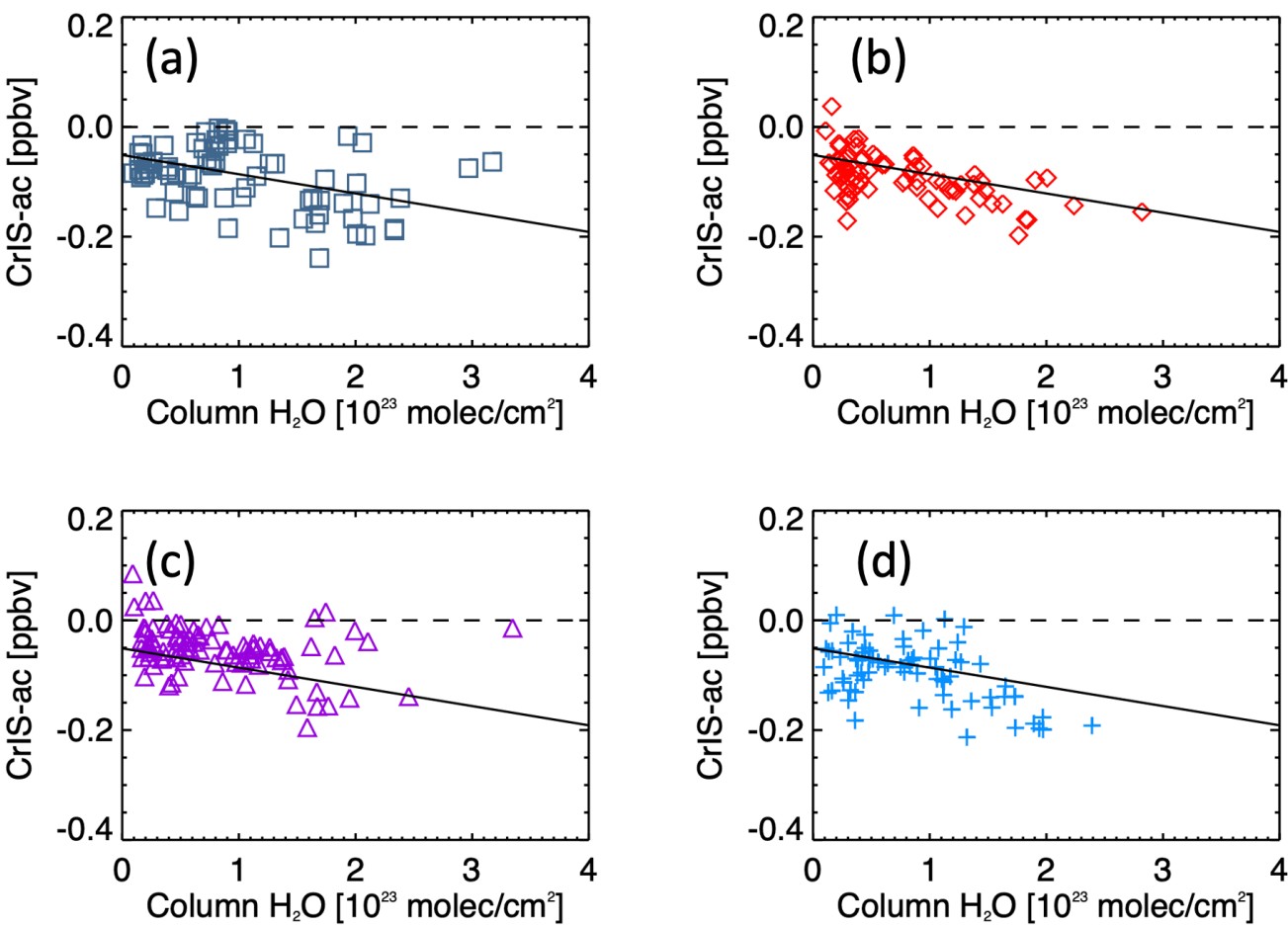

**Figure 9: Scatter plots of CrIS – aircraft PAN differences against column water vapor for (a) ATom 1, (b) ATom 2, (c) ATom 3 and**
**(d) ATom 4. The solid black line, which is the same on all panels, shows the result of a linear fit to the data for all four campaign**
**time periods.**
