# Peer review of "Satellite measurements of peroxyacetyl nitrate from the Cross-Track Infrared Sounder: Comparison with ATom aircraft measurements"

_Atmospheric Measurement Techniques, 2021_

## Author Comment (AC1)

*We would like to thank the reviewers for their well-considered and constructive comments and questions, which have definitely helped us to improve this paper. We have made extensive updates to the paper as a consequence of the comments. Our responses to reviewer comments are in italicized text below.*

**Reviewer 1**

**General comments**

The study of Payne et al. (2021) presents new PAN retrievals from CrIS obtained by optimal estimation, which fits perfectly in the scope of AMT. The retrieval methodology is clearly explained and the validation against aircraft measurements is carefully led. This study demonstrates the ability of CrIS to measure PAN and capture its variability even in background conditions.

Therefore, I recommend the publication of this paper in AMT. However, I have a few comments and questions listed below.

**Specific comments:**

**- Abstract and Section 5 ("Discussion and Conclusion"): "This bias does not appear to show a dependence on latitude or season."**

This statement does not look so clear to me, looking at Fig.6. Without any dependence, we would expect a randomly distributed differences (around the mean bias of -0.08 ppbv), while it looks like in Atom1 and 3, the biases are larger for latitudes 5-40°N, and in Atom2 and 4, larger for about latitudes 30°S-10°N (although less clear here, due to smaller sampling). This could have been due to a systematic bias effect and not a latitudinal effect if CrIS was showing a systematic bias with the aircraft, but the comparisons show a slope of 0.99, so only a constant absolute bias is expected (and furthermore Fig.3 is not showing maximum of PAN at these latitudes, except maybe for Atom2). Can the authors explain more why they assume that there is no dependence on latitude? What could be the reason of the larger bias in Atom1/3 in 5-40°N?

*On reflection, we agree with the reviewer. We have performed additional analysis to examine sources of systematic error and have updated the paper accordingly.*

*We have now expanded the y-scale on Figure 6 to make it easier to see the variations that the reviewer refers to. We also found a small mistake in the code used to generate the scatter plot and calculate the aircraft/PAN slope. The intention had been to only use cases where there were at least 5 CrIS matches for a given aircraft profile, but this had not previously been applied in the calculation of the slope. This update results in some very small changes to the numbers in Table 1. In addition, we performed additional work to examine the robustness of the slope calculation (see response to comment below). As part of that, we weighted the satellite PAN*

*values according to the standard deviation of the PAN within the CrIS matches for each aircraft profile. After these changes, we find a slope of 1.07+/-0.06.*

*We find that the bias depends on column water vapor. We have examined the spectral residuals over the 760-860 cm-1 region before the PAN retrieval step. We find that there is an overall positive offset in the radiance residuals between 760 and 775 cm-1 before the PAN is retrieved. This can in fact be seen in the example shown in Figure 4. This offset tends to be larger for cases with larger column water vapor. This could indicate either a systematic bias in the retrieved water vapor that is fitted in steps before the PAN step, or a systematic bias in some aspect of the water vapor spectroscopy in this specific spectral region. Note that we do exclude cases where the quality flag for the water vapor retrieval from a previous step was "bad". (This was done for the original submission, but this information was not previously stated explicitly – we have now added a statement in Section 4 to state this explicitly.) We speculate that if the true atmospheric PAN were uniform with latitude, with all values at the low end of the range (say < 0.1 ppbv), then we would see a retrieval bias that would follow the latitudinal structure of water vapor. At very low PAN values, our retrieval is attempting to compensate for the offset by underestimating the PAN. However, at higher PAN values, the shape of the PAN feature is stronger, allowing the retrieval to better distinguish the shape of PAN from an offset and so the bias is reduced. This is consistent with a satellite/aircraft slope that is greater than 1.0, which is what we now find.  We have added a figure (Fig 9 in updated manuscript) to show the dependence of the bias on column water vapor, and have updated Section 4 to include further discussion of this issue.*

*We have also now included some discussion of an approach for empirical correction of this bias, as requested by the other reviewer, and updated the original Figure 7 (Figure 8 in updated manuscript) to show the impact of such a bias correction on the aircraft/CrIS scatter plot.*

*Ideally, we'd have an approach in place within the algorithm to somehow correct for the radiance residual offset before fitting the PAN. In the version of the algorithm for which data are currently publicly available, we do not have such an approach in place. We have now added comments to this effect to the discussions section.*

**- Section 3.1 Retrieval algorithm and strategy**

You mentioned in Sect. 4, that you also used Rodgers 2000 for deriving estimated uncertainties (noise, model parameters, ...), and that the theoretical error values are too small by a factor or 2-3. You are talking about only the random part I guess. Do you have an estimation of the systematic error budget as well (spectroscopy, …)? Providing here in Sec. 3.1 a "theoretical" random and systematic uncertainty budget for individual PAN retrievals would be helpful for all the users (in addition to the estimated value for the precision provided in this study by the std of the comparisons).

*The reviewer is correct. When we said the theoretical error values are too small by a factor of 2-3 we were indeed only talking about the random part. We have updated the text to make this clear and to also make it clear that the random part is all that is reported in the current version of the CrIS PAN product files. However, we definitely agree that an uncertainty budget would be helpful for users.*

*We have updated the paper to include discussion of systematic error terms. The additional discussion includes spectroscopic uncertainty, interfering species and land surface emissivity.*

(1) *Spectroscopic uncertainties. We now include the following material: "We used PAN cross sections taken from the work of Allen et al. (2005a; 2005b), which cover a temperature range of 250-295 K. Allen et al. (2005a) cite uncertainties in the integrated band intensity of around 7 % for the band used here, but this does not include the extrapolation error below 250 K. Following the logic of Tereszchuk et al. (2013), we assume a value of ~13 % for the spectroscopic error."*

(2) *Interfering species. To the extent possible, the PAN microwindows were selected to avoid interfering species, resulting in the small size of the microwindows shown in red in Figure 4. However, as both reviewers points out, we can still expect some contribution of interfering species to the overall error budget. Water vapor, carbon dioxide (relied on for temperature retrievals) and ozone all have lines in the region of interest of the 790 cm-1 PAN feature. After those three, the next biggest interferent contribution is from CCl4. The interfering species can be split broadly into two categories: Those that are retrieved (which can be thought of as systematic errors with pseudo-random impacts) and those that are not (which are simply systematic).*

   a. *In the retrievals that are run routinely for the TROPESS project, temperature, water vapor, and ozone are all retrieved in previous steps before the PAN step is attempted. For the purposes of estimating systematic error contributions, we performed additional runs for this set where the estimated retrieval errors for temperature, water vapor and ozone are propagated into the PAN retrieval. When the retrievals are run in this way, the mean estimated observation error for PAN is 0.06 ppbv, which is closer to, but still somewhat lower than ,the standard deviation of the CrIS/aircraft differences. We find that the propagated temperature error is the dominant piece of this, followed by water vapor and that the error associated with ozone is negligible. Note that the propagation of water vapor retrieval error, which can be thought of as a "pseudo-random" systematic error, is separate from the "pure" systematic bias that is dependent on water vapor column that is discussed in response to the first comment above. Section XX has been updated to include this information.*

   b. *The current algorithm uses a CCl4 climatology that is static in time and was constructed around the time of the launch of the Aura satellite (2004). Ground-based measurements from the NOAA Global Monitoring Laboratory indicate that CCl4 has decreased by around 20 % between 2004 and the current day. There are plans to update this (and other) climatology(ies) in future algorithm versions to account for known time dependence. We performed runs where the CCl4 climatology was scaled by 0.8 in order to get a sense of the magnitude of systematic error associated with mis-specification of CCl4. The mean impact on the PAN retrievals for the set of ATom match-ups was only 0.01 ppbv. CCL4 is expected to continue to decrease over time, so if not addressed, this would become a bigger issue as time goes on.*

(3) *Land surface emissivity: For retrievals over ocean, which make up the overwhelming majority of the ATom coincidences discussed here, we assume that the surface emissivity is well specified. Over land, we use a surface emissivity database and do not attempt to fit*

*for surface emissivity within the PAN windows. The 790 cm-1 PAN feature coincides with a silicate feature, which is not always well characterized in land surface emissivity database. This can lead to problems with retrievals over sandy/rocky surfaces. Following Juncosa-Calahorrano et al. (2021), we do screen out cases where the surface emissivity database indicates a strong silicate feature. However, the choice of thresholds for screening will always leave some associated uncertainty. Validation over a wide range of surface types would be desirable for the future.*

| Index | Uncertainty | Nature | Estimated magnitude for a case with 0.2 ppbv in mid troposphere | |
|---|---|---|---|---|
| | | | *ppbv* | *%* |
| 1 | Instrument noise | random | 0.03 | 15 % |
| 2 | Total calculated observation error, including propagation of temperature, water vapor, ozone retrieval errors | Pseudo-random | 0.06 | 30 % |
| 3 | Spectroscopic uncertainty | Systematic | 0.02 | 13 % |
| 4 | CCl4 | Systematic | -0.01 or less | -5 % or less |
| | Aggregate of estimated uncertainties for retrieval over ocean | | 0.09 | 45 % |

**P5, l-130-135:** Do I understand well that PAN is retrieved in the Red windows, but that the previous steps (temperature, H2O, …) are fitted in the large window 760-860 cm[-1]? Because I guess fixing H2O is important if the strong H2O line is not included in the PAN fit. Maybe clarify.

*Yes, PAN is retrieved in the red windows. The previous parameters are fitted in different windows. Example residuals for the large 760-860 cm-1 window were calculated off-line for the purposes of PAN algorithm development, and are show here to provide context for the small red PAN windows. We have added a sentence to clarify.*

**Fig.5**: It's nice to see what is measured by CrIS within one day. I guess OE retrievals take too much time to obtain global seasonal maps, and so you have to focus on short periods and collocated regions (such as for these aircrafts campaigns)? It would have been nice to see seasonal global distribution of PAN from CrIS: how long would it take to process, for e.g. one year of data?

*Yes, there are limitations in processing speed for this algorithm, and so yes, for the TROPESS project under which these CrIS PAN retrievals have been developed and are being run, the initial focus has been on selected subsets of data. The CrIS instrument measures millions of spectra per day. The existing plan under the TROPESS project, a five-year project that started in 2020, is to process 30,000 obs/day from 2016 to 2025.*

*The CrIS record, starting with the instrument on the S-NPP satellite and continuing with the JPSS series, starts in late 2011 and will continue into the 2030s and beyond. Perhaps at some future time there will be an opportunity to create a PAN record for the full CrIS record (all spectra, all years).*

**-Section 3.2 Initial guess and a priori constraints: "The initial guess profile values for these CrIS PAN retrievals are set to a vanishingly small number."**

Do I understand from that sentence that the a priori profile (used in Eq. 1) are different than the initial guess profiles? If yes, why not using a priori profiles as initial guess?

*Yes, these are different. The rationale for this choice lies somewhat with the history of the progression of the PAN algorithm development. It was convenient to set the initial guess to something tiny in order to work with and check wide-filter residuals for visible PAN signatures while developing the PAN algorithm. The solutions for the PAN retrievals do not depend on the choice of initial guess, so it does not matter whether we use the initial guess set to the prior or to the tiny value.*

**- Section 3.2 (should be 3.3) Vertical sensitivity**

Could you give the mean DOFS and std for, for example, a typical day as in Fig. 5. And if the std is large, a little bit more information on which conditions give the best DOFS (e.g. enhanced PAN values, but maybe other factors are influencing the DOFS)? Also you could provide the mean and std DOFS for the PAN retrievals used in the comparisons, which are reflecting more "background conditions". Also, how much do you lose in % of DOFS by taking 800-300hPa information instead of total columns?

*We have now added a figure that provides more information on the total and partial DOFS, as well as additional discussion in Section 3.2 (vertical sensitivity). The largest DOFS are for cloud-free conditions. The DOFS for these retrievals depend mainly on the nature of the cloud field (cloud optical depth and cloud height). Clouds that are optically thick will tend to block sensitivity to the atmosphere below. Since the sensitivity is mainly in the mid-troposphere, optically thick clouds near the surface have little impact on the DOFS, but optically thick clouds with high cloud top pressures have a large impact.*

**Section 4 Results**

**P7, l. 198-199: "(most of the aircraft measurements are at the low end of the range)":** What do you mean? Not clear for me.

*We had intended to make the point that the mean difference between datasets is a small absolute number in ppbv. The reason the mean difference is so small in absolute terms is that the bulk of the ATom measurements are for very low PAN. On reflection, we have decided to remove the statement about the mean difference between aircraft datasets, since this is less important than the information about the linear fit between the two sets of aircraft measurements.*

**P7, l. 203-205:** The correlation with a priori is already good. Especially with the GT_CIMS, where the improvement with retrieved data could appear limited (0.64 compared to 0.53 for a priori). But looking at Fig. 7, it might be due mainly to the isolated point at 0.46 ppbv. Maybe use robust statistics to derive the correlation and the slope and reduce the effect of a single point (it might reduce correlation with the a priori while keeping good correlation for retrieved values and then strengthen the added value of your retrievals). The slope with retrieved values would also be more accurate by using robust statistics.

*The correlation with the a priori is indeed already pretty good. The reviewer's comments about robust statistics led us to go back and examine this carefully. We made two important updates since the submitted version of the paper. One was that we fixed a mistake where we had previously failed to exclude cases where there were less than 5 CrIS match-ups to a given aircraft profile in our calculation of the linear fit. The second was that we updated the analysis to weight the CrIS averaged points according to the standard deviation within each set of CrIS match-ups to a given aircraft profile. In order to address the question about robust statistics, we performed a bootstrapping analysis with 10,000 samples. After the two updates described above, we find that the linear fit returns a slope of 1.07+/-0.06, with an R^2 coefficient of 0.62. The bootstrapping analysis returns the same numbers (consistent to the second decimal place), and we can conclude that the numbers are robust. We have updated the relevant figure and included the estimated error on the slope along with some additional discussion in the text.*

**Section 5 ("Discussion and Conclusion")**

**P8, l. 234-235:** While the comparisons are made very carefully and the conclusion "The results … demonstrate the ability of the CrIS PAN retrievals to capture variation in the "background" PAN values observed over remote ocean regions from Atom" is certainly reached, I would be less assertive concerning the statement "Based on this study, we expect this bias to apply to all parts of the world".

Indeed, while we see in Fig. 2, that the aircraft measurements often sample PAN levels up to 0.48 ppb, the comparisons with CrIS is limited to 0.32-0.34 ppb (with the exception of only one coincidence), I guess due to collocation. When we look at Fig. 5, where the PAN retrievals can reach 0.90 ppb in some regions, I think that the present study is not covering all the gradient of PAN concentrations to make this statement. Validation at high concentration conditions should be done before concluding that the bias would be the same. It is quite usual to have different satellite biases over clean and over high concentration sites (e.g. TROPOMI HCHO, Vigouroux et al., 2020; TROPOMI tropospheric NO2 Verhoelst et al., 2021; …). Of course, with the interesting approach used here (validating 800-300 hPa, and not tropospheric column), the results might be different but I think it's worth looking at additional validation at high concentration conditions before concluding that the bias will apply to all parts of the world.

*This is a fair comment. We agree that validation over high PAN conditions would be desirable. We have removed the statement about the bias applying to all parts of the world and replaced with a statement about how we would ideally have validation in a wider range of PAN conditions, and over a variety of land surface types.*

**Minor or technical comments:**

References Payne et al, 2014, 2017 are missing.

*These have now been added.*

Figure 1, legend: NEdT is not defined
*Caption has been updated.*

---

## Author Comment (AC2)

*We would like to thank the reviewers for their well-considered and constructive comments and questions, which have definitely helped us to improve this paper. We have made extensive updates to the paper as a consequence of the comments. Our responses to reviewer comments are in italicized text below.*

**Reviewer 2**

GENERAL COMMENTS

==================

The paper describes a new level 2 data set for the CrIS instrument. The retrieval process is briefly described and an in depth comparison to in situ data gathered by the ATom campaign(s) is given. Due to the number of CrIS instruments in orbit and in planning, this is an important data product. The paper identifies a strong bias of the provided data, which is larger than the assumed uncertainty. A H2O-VMR-based bias correction is suggested in the User Data documentation, but not discussed in the paper itself. The paper should address the bias more explicitly and discuss causes and corrections. Ideally, the root causes for the bias should be identified and the data product improved.

I recommend publication after revising the paper to discuss these points in detail and answering the other comments below.

*We thank the reviewer for the detailed and insightful review and for the positive statement about the importance of this data product.*

*We appreciate the reviewer taking the time to check the user guide against the information presented in this manuscript. We agree that the paper ought to reflect the information in the user guide.*

*Updates to the user guide are underway. In the time between when the first version of the user guide was made available and when this paper was submitted, we did some more thinking about problems associated with water vapor interference in the CrIS PAN retrievals. As we have discussed in the paper, water vapor is a strong interferent in the spectral region used for the CrIS PAN retrievals, and is retrieved separately in a step before the PAN retrieval step. However, the master quality flag in the PAN products that are in the forward stream CrIS PAN dataset (cite doi, access date) does not include a check on the quality of the water vapor retrieval from that previous step. We find that a large number of cases in the Tropics with "bad" quality for the water vapor step were associated with strongly negative PAN retrievals. Those strongly negative PAN retrievals in the Tropics that were having a large impact on the H2O-VMR-based bias correction described in that initial version of the user guide. If we screen out cases where the water vapor retrieval step fails quality control, then the H2O-VMR-based bias correction (that had been based on "bad" cases) is less severe. In the version of the dataset that is now currently available, we recommend that the user use the H2O product information for screening the PAN. In future algorithm updates, it will make sense to directly account for the quality flag from the H2O step in the quality control for the PAN step. In the revised version of the*

*manuscript, we now include information on an updated formula for bias correction and we are working to get a revised version of the user guide posted with the TROPESS PAN products.*

MAJOR COMMENTS

===============

line 132
* * *
The paper identifies a bias of -100 pptv in the derived data, which is larger than the supplied uncertainty in the data (80 pptv) derived from the standard deviation computed from differences to in situ measurements. This suggests that the bias is real and significant, particularly for non-polluted airmasses outside of plumes. The employed spectral region is full of emission signatures of a wide range of trace gasses. It seems as, e.g., CCl4 could still have an effect, but also other CFCs, or ClONO2 emit in this region. While the strong H2O emission line at 785 has been avoided, weaker lines are certainly present in the left window. The User Guide for the data even provides a bias correction formula depending on water vapour.
I question the usefulness of the data set in the current state.

1) Why was the obvious and *astonishingly* stable bias not corrected in the data set?

*We have now extensively updated the manuscript to include discussion of this water vapor dependent bias as well as discussion of additional possible sources of systematic errors in the PAN products. Please see the response to reviewer 1 for discussion of the water vapor dependent bias. We have now included discussion of the bias correction formula in the paper.*

*The water vapor-dependent bias discussed above is by far the dominant source of purely systematic bias. The impact of uncertainties in specification of CCl4 is small (~0.01 ppbv), and the optical depth contributions of the CFCs and CLONO2 are smaller than CCl4. Note that we have now also included discussion of the propagation of estimated retrieval errors in temperature, water vapor and ozone (which are retrieved in previous steps) into the PAN retrieval. These can be regarded as "pseudo-random" contributions to the observation error, and we find that including these terms can at least partially account for the discrepancy between the observation error from instrument noise and the standard deviation of the differences between satellite and aircraft values (theoretical vs empirical).*

2) Why was the retrieval not improved to the point, where no bias correction is necessary?

*We strongly agree with the reviewer that the ideal scenario would be to address the root causes of any biases in remotely-sensed products and eliminate them completely. However, this idealized goal may or may not be attainable. While further improvements to future versions of the product are desirable, the current situation is that version of this product is available to the public and the CrIS PAN data are already being utilized for science studies. This paper describes the product and documents the observed bias.*

3) Why was the bias correction formula of the User guide not mentioned or applied for the comparison?

*We agree with the reviewer that the bias correction referred to in the User Guide should be discussed and applied here for the comparison. This has now been addressed in Section 4, in what are now Figures 9 and 8 and in the abstract.*

Figure 2
* * *
High PAN VMRs occur often at higher tropospheric altitudes (particularly due to the longer lifetime at colder temperatures) close to the tropopause. The used aircraft data rarely go above 12km. Biomass burning plumes reach higher than 12km, particularly in the tropics. The given altitude range of 800hPa to 300hPa is key here, as 300hPa corresponds roughly to 10km.

How does this limited altitude range affect the accuracy of estimating total PAN in the UTLS?

*If we understand correctly, the reviewer is asking about the component of the error budget associated with what we have assumed for the profile above the top of the aircraft measurements. This is a good point, and something that was not discussed in the initial manuscript version. As stated in the initial version, GEOS-Chem model output for runs specific to the time period was appended above the uppermost and below the lowermost altitudes spanned by the ATom PAN profiles. Due to the vertical sensitivity of the CrIS PAN retrieval, the assumption about what to append above the top of the aircraft profile is far more important than what is appended at the bottom, (provided what we append at the bottom is reasonable!) If we did not have these targeted model runs available, an alternative crude approach could have been to simply append the retrieval prior to the top of the aircraft profiles. The difference between these two approaches provides some estimate of the uncertainty associated with the assumed profile above the top of the aircraft measurement. We find a 20 % reduction in the aircraft/satellite slope between the case where we append the prior and the case where we append the dedicated GEOS-Chem runs. We can think of this as a pessimistic estimate of the error associated with the assumption of the profile above the top of the aircraft profiles. We have now included this information in Section 4.*

Why is the instrument not sensititve (at all? enough?) to high PAN VMRs closer to the tropopause? Is this related to the low temperature at this altitudes?

*CrIS is sensitive to PAN throughout the atmospheric column. We have now added a figure that shows a representative averaging kernel and additional discussion of vertical sensitivity in Section 3.2. Please also see responses to Reviewer 1's question about the DOFS.*

SPECIFIC COMMENTS

==================

line 135
* * *
Particularly in the face of the discovered systematic error, a discussion on the sensitivity of the retrieved PAN VMRs on the previously derived quantities (i.e. the 'b' vector) might be interesting. It is mentioned that the retrieval processor under-estimates the "observation error", without detailing what exactly this entails. Often this only contains - for practical reasons - an estimate of the noise induced error, not the systematic errors. How does the identified systematic bias relate to the error diagnostics for systematic (b-related) errors?

*This is a good suggestion. In response to comments from both reviewers, we have now made extensive updates to the manuscript to include discussion of systematic errors.*

line 165
* * *
Please show a (representative set of) averaging kernels to show the region of sensitivity.

*We have now added a figure that shows an example averaging kernel and added some discussion of this figure to the text.*

Figure 4
* * *
The residual shows structure beyond the noise level (blue lines). The caption indicates that this spectrum was computed with a zero PAN profile. Please show both a spectrum with the derived PAN profile and with a zero profile to show the improvement and PAN signal as well as quality of fit of the used spectra (similar to Glatthor et al., 2007)

*Note that the figure in question shows sample residuals over a large range (760-860 cm-1), but that the PAN windows are pretty small. PAN is only fitted within the small red windows. For retrieval development purposes, we had set up runs where the "pre-PAN" state information (including retrieved temperature, water vapor and ozone profiles) is run through the forward model for the 760-860 cm-1 range to generate these wide filter residuals. We could, in principle, update our system to run a "post-PAN" step to generate the wide filter residuals after the PAN retrieval. However, this would take some effort. We think that the figure shown is sufficient to show the expected PAN signal and so we would rather leave this figure as it is.*

*We also note that one benefit of the figure as it stands is that it does show an example of the radiance offset issue that exists before the PAN step that leads to the water vapor dependent bias.*

Figure 5
* * *
The paper identifies a low bias of 100 pptv causing many VMRs to be negative as shown in Fig. 7. Figure 5 shows only positive VMRs. Please explain the discrepancy.

*This was an oversight that has now been corrected. The data used to generate Figure 5 did include negative values, but we had set the color scale to bottom out at zero. The black region at the bottom of the original bar was misleading. We have updated the color scale to show the regions where PAN remains systematically negative, even after the water vapor dependent bias correction has been applied, and we have updated the caption to make it clear that the color corresponding to the lowest box on the color bar includes points that are more negative than the lowest marked values. We also include an additional figure that shows the actual range of values for this day.*

MINOR REMARKS

==============
line 108
* * *
A big X with a hat was not in (1). Maybe big-hat-x -> hat-x and hat-x-a -> hat_x ?

*Yes. Now fixed. Thank you.*

line 113
* * *
\Delta f should be 'bold'.

*Fixed.*

line 128
* * *
an approximate solution?

*We prefer to leave this the way it is.*

line 134
* * *
CCl_4 (small l)
line 139
*Fixed.*
* * *
It is not clear from the context what the "forward stream" is. The given reference distinguishes a "reanalysis stream" without being clear on the difference. I suppose it has something to do with using (forward) extrapolation of calibration data in contrast to interpolation using (later) data. This is probably a very common term in certain scientific communities. Maybe explain it in a brief sentence.

*The distinction between "forward stream" and "reanalysis" in the TROPESS datasets is that the forward stream represents low latency processing with the most recent algorithm version available at the time of the measurements, meaning that the data version can change in time, whereas the "reanalysis" represents a processing of a long-term dataset with a consistent version of the retrieval algorithm. We have now added some explanation of the forward stream and reanalysis to this paragraph.*